# Reversible Quantum-Dot Cellular Automata-Based Arithmetic Logic Unit

**DOI:** 10.3390/nano13172445

**Published:** 2023-08-29

**Authors:** Mohammed Alharbi, Gerard Edwards, Richard Stocker

**Affiliations:** 1Division of Electronic and Electrical Engineering, School of Engineering, Faculty of Engineering and Technology, Liverpool John Moores University, Liverpool L3 2ET, UK; g.edwards@ljmu.ac.uk; 2Department of Computer Science, Electronics and Electrical Engineering, Faculty of Science and Engineering, University of Chester, Chester CH1 4BJ, UK; r.stocker@chester.ac.uk

**Keywords:** quantum-dot cellular automata (QCA), arithmetic logic unit (ALU), reversible, energy dissipation

## Abstract

Quantum-dot cellular automata (QCA) are a promising nanoscale computing technology that exploits the quantum mechanical tunneling of electrons between quantum dots in a cell and electrostatic interaction between dots in neighboring cells. QCA can achieve higher speed, lower power, and smaller areas than conventional, complementary metal-oxide semiconductor (CMOS) technology. Developing QCA circuits in a logically and physically reversible manner can provide exceptional reductions in energy dissipation. The main challenge is to maintain reversibility down to the physical level. A crucial component of a computer’s central processing unit (CPU) is the arithmetic logic unit (ALU), which executes multiple logical and arithmetic functions on the data processed by the CPU. Current QCA ALU designs are either irreversible or logically reversible; however, they lack physical reversibility, a crucial requirement to increase energy efficiency. This paper shows a new multilayer design for a QCA ALU that can carry out 16 different operations and is both logically and physically reversible. The design is based on reversible majority gates, which are the key building blocks. We use *QCADesigner-E* software to simulate and evaluate energy dissipation. The proposed logically and physically reversible QCA ALU offers an improvement of 88.8% in energy efficiency. Compared to the next most efficient 16-operation QCA ALU, this ALU uses 51% fewer QCA cells and 47% less area.

## 1. Introduction

Heat dissipation in conventional, complementary metal-oxide-semiconductor (CMOS) technology is a major challenge for the design and operation of integrated circuits (ICs). As CMOS technology scales down, the power density and the operating temperature increase, which can degrade the performance, reliability, and lifetime of devices [1,2].

Conventional methods of computing typically include irreversible operations, which result in some input bits of information being erased during the process. In 1961, Landauer [3] proved that irreversible computations cause information loss and involve an amount of heat dissipation of *k_B_T*ln2 per bit erased, where *k_B_* is the Boltzmann constant, and *T* is the temperature. In 1996, Gershenfeld [4] argued that the actual amount of energy dissipated due to information loss is much higher than Landauer’s lower bound. As nanoelectronics circuits and systems decrease in size and become more efficient, their energy dissipation levels approach Landauer’s limit. Therefore, to continue the trend of reducing power consumption and to reach Landauer’s lower bound, unconventional computation methods that allow for reversible logic operations without information loss are needed [5].

Reversible logic operations, which have a one-to-one correspondence between the number of input and output signals, are a promising alternative to conventional irreversible computations that lose information and consequently dissipate heat into the environment. In 1973, Bennett [6] showed that reversible computations can theoretically avoid information loss and achieve zero energy dissipation. Reversible computing is a paradigm of computation that allows computational processes to be reversed in time, recovering previous states of the system. This property is essential for avoiding the increase in physical entropy and the associated energy dissipation that occurs when information is erased irreversibly. However, substantial energy reductions in reversible computing can only be obtained by maintaining reversibility down to the physical level [7]. This means that not only the logical operations but also the physical devices and circuits that implement them must be reversible to avoid energy dissipation.

Quantum-dot cellular automata (QCA) are promising nanotechnology for implementing digital logic circuits that are logically and physically reversible and overcome the drawbacks of conventional CMOS-based very large-scale integration (VLSI) technology, including high power consumption and heat dissipation. In 1993, Lent et al. [8] presented a quantum-dot physical implementation for digital logic circuits. Using field-coupled nanotechnology (FCN), quantum dots encode information as the polarity of the electron orientation, which is coupled to neighboring cells via the electrostatic interaction [9]. QCA cells, which are comprised of four quantum dots placed towards the four corners of a square, are the fundamental building blocks of QCA circuits. Each cell has two unbound electrons that are able to tunnel between quantum dots, representing two binary configurations. Because of their electrostatic interactions, the two electrons tend to occupy opposite corners. Figure 1 shows that the QCA cells can be in either of two possible electron configuration states, labeled by the cell polarizations P = −1 and P = +1, which correspond to the binary digits 0 and 1, respectively. Using arrays of quantum dots, QCA encodes binary information and implements specific logic gates according to the layout of the quantum dots. Many researchers have addressed the QCA paradigm as a promising future computer technology as the specific circuit structure and electrostatic interactions among neighboring cells that allow logic functions to be executed [10,11,12].

Every computer’s central processing unit (CPU) contains an arithmetic logic unit (ALU) that performs digital logical and arithmetic operations with binary numbers. Combinational logic circuits are commonly utilized in the process of developing ALU circuits. Although reversible circuits have been demonstrated to improve energy efficiency, the preponderance of published QCA ALU designs is irreversible. Many researchers have recently started studying reversible QCA ALU architectures [13,14,15,16,17,18]. However, the information loss that occurs due to reversibility at the physical layout level has not been considered in these studies. To design reversible ALUs, these studies utilized either widely used logically reversible gates, such as the 3 × 3 Fredkin gate or the 2 × 2 Feynman gate, or newly proposed logically reversible gates [13,14,15,16,17,18]. The netlist of these designs has an equal number of input and output pins, but this is not enough to make them physically reversible to produce ultralow-energy dissipation ALUs. The reason for this is that the internal majority gates used to build these ALUs are not reversible; each internal majority gate has a different number of input and output pins. In 2020, Torres et al. [19] presented the first implementation of the logically and physically reversible design principle for designing and simulating a QCA half-adder circuit. They developed it using the *QCADesigner-E* tool, a QCA circuit design and simulation tool that Torres et al. invented [20]. Using *QCADesigner-E*, the energy dissipation of the QCA half-adder circuit was calculated. The results showed that combinational QCA circuits that are both logically and physically reversible can operate with energy dissipation values lower than Landauer’s limit. Later, researchers implemented this technique to develop more sophisticated combinational QCA logic circuits, as well as sequential QCA circuits, characterized by feedback loops [21,22]. In this research, we introduce the first multilayer reversible QCA ALU built using the logically and physically reversible QCA design technique. This ALU design relies mainly on the majority gate as the core building block and uses the universal, standard, and efficient (USE) clocking scheme to synchronize data propagation. Our simulation results confirm that the logically and physically reversible design method can yield an ultralow energy dissipation QCA ALU. The remainder of this paper is organized as follows: In Section 2, QCA clocking schemes are reviewed. In Section 3, the logically and physically reversible design method is defined. In Section 4, the multilayer logically and physically reversible QCA ALU design and simulation setup are described. Then, the simulation results are discussed in Section 5, and the conclusions of the study are stated in Section 6.

## 2. QCA Clocking Schemes

To ensure proper data transfer and operation in logic circuits, clocking control plays a vital role in coordinating data flow. For QCA, an external clock is needed to alter the tunneling barrier strength between the QCA cells and achieve clocking control. Various timing and clocking methods have been proposed to regulate the transmission of information via QCA circuits.

In 1997, Lent and Tougaw devised adiabatic switching as a way to control the timing, deal with metastability problems, and facilitate pipelined creation for QCA circuits [9]. This clocking method divides the QCA array into clusters of cells known as clock zones, providing the advantages of multiphase clocking and pipelining. Using the clock zone structure, a cluster of QCA cells can perform a computation, freeze its states, and finally feed the results into the next clock zone as inputs. Allowing the QCA wire length to grow can increase the risk that cells will not switch accurately due to thermodynamic constraints; therefore, dividing the wire into zones is analogous to breaking it into several smaller wires [23].

The four phases of the adiabatic pipelining cycle are depicted in simplified form in Figure 2. Each outline box represents a multicellular clock phase. Each cell in a clock phase uses the same gate to control inter-dot barriers. In each box, the cell on the left depicts the state of the cells at the commencement of this clock phase, while the cell on the right depicts the state of the cells at the end of this clock phase. The single cells depicted can represent a subarray of QCA cells. At the beginning of the first phase, referred to as the *switch phase*, the cells are unpolarized and have low barriers. However, as this phase progresses and computation is performed, the barriers increase, and the cells become polarized to correspond with the computation function. This phase ends with substantial barriers that prohibit tunneling and fixed cell states. The second phase is termed the *hold phase*, where the barriers remain at their current height. The third phase is the *release phase*, where the barriers are lowered, and the cells become depolarized. The *relax phase* is the fourth and final phase of the clock. This phase maintains the unpolarized nature of the cells by keeping cell barriers low. After the fourth phase, the clock system repolarizes and reverts to the first phase, beginning a new cycle.

In the adiabatic switching method, the input states are switched gradually, while the interdot barriers of the cells are changed, at the same time, across the whole array. This keeps the system in an instantaneous ground state. Furthermore, the data can be synchronized so that no signal reaches a logic gate and propagates before any other inputs reach the gate. Having these features guarantees that QCA circuits will function properly. However, the implementation of this one-dimensional adiabatic switching scheme faces a number of obstacles. These obstacles include the difference in the lengths of the wires, the clock zone capacities, and the number of cells between the different clock zones, which may preclude the construction of feedback paths and produce an unused area [24].

In 2007, Vankamamidi et al. [25] proposed a two-dimensional QCA timing method. The two-dimensional QCA clocking method can achieve higher performance and lower power consumption than a one-dimensional QCA clocking method by exploiting the spatial and temporal parallelism of QCA circuits. This clocking scheme takes zone size into account and comprises a grid of square zones that are equal in size, thus preventing thermodynamic effects on QCA circuits. The overhead of feedback channels, however, remains a major challenge [24]. In advanced QCA circuits, long lines between timing zones have a negative effect, leading to higher delays and sensitivity to thermal fluctuations [24].

Campos et al. developed the unified, standard, and efficient (USE) timing method in 2016 [24]. The adaptability of the USE timing method enables it to satisfy the requirements of the QCA circuits design, which include the implementation of feedback channels with small or large loops, the standardization of cell libraries, and the facilitation of routing simplicity. Figure 3 shows the USE clocking system, which consists of four time zones numbered from 1 to 4. These four time zones constitute one complete clock cycle. Data flows between the QCA cells in neighboring clock zones are shown here as squares. Each square contains a cluster of five-by-five QCA cells representing a distinct time zone.

To balance the speed of data transmission and the arrival time of data for each logic gate in the circuit, clock synchronization is an essential metric [26]. The differentiation between local and global synchronization must be carefully considered when evaluating QCA circuits. *Local synchronization* necessitates that data transmission be restricted only between cells in clock zones with consecutive numbers. *Global synchronization* ensures that new data are transmitted to the inputs of the circuit during every clock cycle; thus, the inputs of all gates are synchronized for at least one clock cycle prior to the arrival of new data. Most researchers emphasize that local synchronization is an essential requirement to include when developing QCA circuits [26,27,28,29]. However, the conclusion of global synchronization research is contradictory. In spite of numerous assertions highlighting the importance of global synchronization [27,28], some studies argue that global synchronization is not a necessary requirement for QCA circuits [29].

The real clocking concept is essential for the design of QCA circuits because it can substantially decrease manufacturing costs and facilitate the physical implementation of QCA circuits. Either the pipeline-style [25] or the dynamic-style [24,29] can be used to include the real clocking concept into the QCA clocking system. For complex circuits based on five-input majority gates, the real clocking strategy, with efficient clustering and placement, has been recently developed [30]. Generally, QCA circuits utilizing majority gates with more than three inputs benefit significantly from the real clocking technique [30].

In this study, a multilayer QCA ALU circuit design that is both logically and physically reversible is proposed. To prevent getting stuck in a metastable state and to precisely control the flow of information between the QCA cells, the USE clocking method was employed. The tile-based USE clocking method facilitates routing, the construction of clock zones with uniform, standard, and bounded forms, and the creation of feedback loops with both big and small loops. In addition, USE clocking circuitry may be realized using current integrated circuit (IC) design and production capabilities. Clock synchronization, both locally and globally, is crucial for the design of complex QCA circuits. Clock synchronization ensures that data arrives at the appropriate time for the next stage of the circuit and that the speed of data propagation is balanced [31]. If there is no control over clock synchronization, the subsequent stage may generate inaccurate bits, leading to incorrect data transfer. The ALU proposed in this study transmits information between clock zones numbered sequentially. Each logic gate receives its input data within the same clock cycle, i.e., within four clock phases. This guarantees that the proposed ALU design in this study is both locally and globally synchronized and should generate correct computation results.

## 3. Logically and Physically Reversible Design Methodology

In QCA circuits, the main source of energy dissipation is the majority gate [32]. The conventional QCA majority gate is irreversible with three inputs and a single output, as depicted in Figure 4. Therefore, it has energy dissipation into the environment associated with information loss. Figure 4a depicts the logical design (schematic), whereas Figure 4b depicts the physical design (layout) of a standard irreversible majority gate.

To develop a highly energy-efficient QCA ALU in this study, a fully reversible majority gate was developed, which is the key component in our design. The fully reversible majority gate makes copies of the input data, resulting in an equal number of binary inputs and outputs. Figure 5 depicts the proposed fully reversible majority gate. It has three inputs and three outputs due to the copying of the incoming data. This reversible design strategy, thus, prevents data loss and, consequently, ensures that no energy is lost to the environment. Figure 5a illustrates the logical design (schematic), while Figure 5b illustrates the physical design (layout) of the fully reversible QCA majority gate design.

In this study, we used a two-stage design methodology to develop the proposed logically and physically reversible QCA ALU components, as shown in Figure 6. This method used the proposed fully reversible majority gate, presented in Figure 5, as the main building block.

*The first stage* was to build the ALU components at the logical level (synthesis), which started with defining and generating the netlist of each circuit used to build the ALU. Then, the behavioral description of the circuit outlines the input and output relations. Simulations are ultimately employed, using *Logisim 2.7.1 software*, to validate each circuit’s synthesis. If the simulation produces unexpected results, the output is considered a *failed* result. So, we must go back and modify the circuit netlist and examine the circuit’s input/output relations. When the simulation yields accurate results, this indicates that the circuit synthesis is correct, and the output is deemed a *pass* result, allowing us to proceed to the second stage.

*The second stage* was to create the physical level (layout) of the ALU components based on QCA interconnected devices. The physical level represents the transition from synthesis to a QCA layout, which is the physical representation of the circuits. This process starts with gate placement, identifying the pins’ locations, and routing for each used circuit to build the proposed reversible QCA ALU. Next, layout verification is performed to validate that the layout reflects the circuit synthesis design. Finally, post-layout simulation is performed to validate both the circuit performance and reliability. For this purpose, the widely used QCA technology-based computer-aided design (TCAD) tool *QCADesigner* was employed in this study. *QCADesigner* implements the coherence vector simulation engine (CVSE), which incorporates quantum-level microscopic physical modeling for the operation of the QCA cells [33]. *QCADesigner* is used to evaluate the QCA circuit’s performance, area cost, latency, and reliability. The output is regarded as *fail* if the simulation produces unexpected outcomes. In this case, we must go over gate placement, the pins’ locations, routing, and layout verification. When layout simulation yields correct output, i.e., *pass* results, this indicates that the circuit design has been deemed completed successfully and is ready to proceed to the energy dissipation simulation.

To simulate the energy dissipation of the proposed physically and logically reversible QCA ALU design, the *QCADesigner-E* software (Version 1) application was utilized. *QCADesigner-E* incorporates a power dissipation extension treatment into the CVSE model of the *QCADesigner* design package and uses a quantum mechanical treatment of QCA cells, a rigorous microscopic approach. The CVSE is a transient analysis with a fixed timestep in which new values for the coherence vector components and the tunneling energy are calculated at each iteration step. To be specific, the evolution of the coherence vector λ is determined by solving the differential equations that represent the evolution of the quantum mechanical density matrix using an iterative fixed timestep approach [20]. The cell polarization is given by minus the z component coherence vector; the kink energy of a pair of cells is a constant value based on the electrostatic interaction of the charges of both cells. Each QCA circuit design requires a specific set of technological and simulation parameters (see Table 1), such as the time interval for each iteration (*T_step_*). In the present research, the used time interval (*T_step_*) was 0.1 *τ =* 0.1 *fs*, where *τ* denotes the relaxation time for the dissipation [20]. A small enough timestep is essential to reduce simulation errors and obtain accurate results. The simulation errors, when using this time step, are considered an acceptable numerical energy conservation violation if they are less than 5% [20]. Table 1 outlines all the technology and simulation parameters that were utilized in this study.

Dealing with wire junctions is one of the greatest challenges in the development of digital logic circuits. In the present research, the multilayer technique, proposed by Bajec and Pecar [34], is used to address the wire crossing issue for designing the logically and physically reversible QCA ALU. This method uses three distinct layers to solve the wire junction issue [35], as depicted in Figure 7. It is important to note that the multilayer approach produces more reliable circuits [36].

## 4. Proposed Logically and Physically Reversible QCA ALU Design

The ALU is a crucial component of the CPU. It can perform various logical and arithmetic operations on the data that enters the CPU. The ALU receives input data from registers, memory, or other sources and outputs the result to another register, memory, or device. The fully logically and physically reversible QCA ALU presented in this study is designed to be ultralow-energy efficient. It was developed using a variety of combinational logic circuits that are designed based on the fully reversible QCA majority gate proposed in Figure 5.

The development process began with the creation of a high-level block diagram. As illustrated in Figure 8, the architecture of the proposed reversible ALU consists of three major components: the logic unit (LU), the arithmetic unit (AU), and the control unit (CU). The LU performs logical operations on data, including AND, NAND, OR, NOR, XOR, XNOR, NOT, and transfer. The AU performs arithmetic operations such as addition, subtraction, multiplication, and division on binary numbers. The CU specifies the type of operation to be conducted, either arithmetic or logic, according to its input S_0_.

The reversible QCA ALU circuit receives two input operands, A and B, and then produces two output values, Output1 and Output2. By reversing Output1 to Output2, the reversible QCA ALU can perform two arithmetic or two logical operations simultaneously, giving a total of 16 operations, including eight logical and eight arithmetic operations, as shown in Table 2. Three select input pins, labeled S0, S1, and S3, are used to determine the operation function of the reversible QCA ALU and which operands to use.

Each block’s synthesis is carried out first and simulated with the objective of validating the circuits’ behavior. Figure 5a depicts the schematic of a fully reversible majority gate, which is used as the basic component for circuit synthesis development. Logical synthesis designs for AU, LU, and CU were developed. The design of logical synthesis includes defining and generating the netlist and input-output relationships of the circuits. At this point, simulation was performed using the *Logisim* software (Version 2.7.1) to validate the performance of each circuit synthesis. Note that in the reversible logic circuit synthesis diagrams, the outputs labeled “cp” are copies of the inputs, while those labeled “g” are garbage outputs.

Figure 9 shows the proposed reversible LU. It consists of two reversible majority gates, an XOR gate, an inverter, a buffer, and a 4-to-1 multiplexer to perform eight logic operations.

Figure 10 illustrates the proposed reversible AU, which consists of a half-adder, a half-subtractor, two reversible majority gates, an inverter, and a 4-to-1 multiplexer and can execute eight arithmetic operations.

Then, the internal components required for constructing the reversible LU and AU were developed. The circuits that compose the proposed reversible LU and AU are the reversible XOR, reversible half-adder, reversible half-subtractor, and 4:1 reversible multiplexer. Each circuit was meticulously designed, and its reliability was proven through simulation.

The synthesis form of the proposed reversible XOR circuit is depicted in Figure 11. This reversible XOR consists of two inverters and three reversible majority gates. Equation (1) denotes the standard Boolean expression as the output of the proposed reversible XOR logic circuit.
(1)$$\mathrm{Output}=\left(A+B\right)\cdot \left(\bar{A}+\bar{B}\right),$$

Four reversible majority gates and two inverters make up the proposed reversible half-adder circuit, as shown in Figure 12. Equation (2) describes the Boolean expressions for this circuit:(2)$$\begin{array}{c} \mathrm{Sum}=\left(A\cdot \bar{B}\right)+\left(\bar{A}\cdot B\right)\\ \mathrm{Carry}=\left(A\cdot B\right), \end{array}$$

Figure 13 is an illustration of the suggested circuit for the reversible half-subtractor, consisting of two inverters and three reversible majority gates. Equation (3) defines the Boolean expression that describes the design output.
(3)$$\begin{array}{c} \mathrm{Diff}=\left(\bar{A}\cdot B\right)+\left(A\cdot \bar{B}\right)\\ \mathrm{Borrow}=\left(\bar{A}\cdot B\right), \end{array}$$

The proposed circuit for the reversible 4-to-1 multiplexer is depicted in Figure 14, comprised of three inverters and nine reversible majority gates. Equation (4) specifies the Boolean equation that defines the design’s output.
(4)$$\mathrm{Output}=\left(\left(\left(\bar{S1}\cdot A\right)+\left(S1\cdot B\right)\right)\cdot S2\right)+\left(\left(\left(\bar{S1}\cdot A\right)+\left(S1\cdot B\right)\right)\cdot \bar{S2}\right),$$

Next, the logical design of the CU was developed. As shown in Figure 15, a reversible 2-to-1 multiplexer functions as a CU for turning on either the AU or the LU to execute an arithmetic or logic operation, respectively. This reversible 2-to-1 multiplexer comprises three reversible majority gates and an inverter. Equation (5) is the Boolean expression for the proposed 2-to-1 multiplexer circuit output.
(5)$$Output=\left(A\cdot \bar{S0}\right)+\left(B\cdot S0\right),$$

The logical synthesis of the circuits was then transformed into a physical layout that could be fabricated on a semiconductor chip. The layout development process involves numerous steps, including partitioning, placement, and routing. By interconnecting a group of QCA cells, a QCA circuit layout has been developed. The layout of the logically and physically reversible QCA majority gate, depicted in Figure 6b, was the basic building block for generating the overall QCA circuit layout.

Initially, we created the layout configurations for the reversible XOR, reversible half-adder, reversible half-subtractor, reversible 2-to-1 multiplexer, and reversible 4-to-1 multiplexer circuits using the layout of the fully reversible majority gate, as illustrated in Figure 16, Figure 17, Figure 18, Figure 19 and Figure 20, respectively. Subsequently, the LU, AU, and CU that make up the multilayer reversible QCA ALU were built using these ingredient reversible QCA circuits. The LU, AU, and CU digital circuit blocks were then connected per Figure 8 to yield the proposed reversible QCA-ALU layout. As with the reversible logic circuit synthesis diagrams, the “cp” labels of the outputs in the reversible QCA circuit layout architectures indicate copies of the input information, and “g” labels mean “garbage” outputs.

As illustrated in Figure 16, the delay time of the proposed reversible QCA XOR gate is eight clock zones (two clock cycles), the occupied area is 0.15 µm^2^, and the number of QCA cells employed is 101.

Figure 17 shows that the proposed reversible QCA half-adder has 12 clock zones (three clock cycles) of delay, occupies 0.27 µm^2^ of area, and requires 156 QCA cells for implementation.

The reversible QCA half-subtractor latency is eight clock zones (two clock cycles), costs 0.15 µm^2^ of area, and its implementation requires 116 QCA cells, as depicted in Figure 18.

Figure 19 demonstrates that the reversible QCA 2-to-1 multiplexer has a delay of four clock zones, which is equivalent to one clock cycle. In addition, its implementation requires 56 QCA cells and 0.09 µm^2^ of area.

The circuit of a reversible QCA 4-to-1 multiplexer was built through the integration of three 2-to-1 multiplexers, as shown in Figure 20. This circuit uses 213 QCA cells, occupies 0.46 µm^2^ of area, and has a delay time of 12 clock zones (three clock cycles).

The proposed reversible QCA LU was created by combining three reversible QCA majority gates, a reversible QCA XOR, and a reversible QCA 4-to-1 multiplexer; see Figure 21. The latency of this reversible QCA LU is 14 clock zones (3.5 clock cycles), the area cost is 0.63 µm^2^, and the required QCA cells for implementation are 380.

Figure 22 depicts the integration of two reversible QCA majority gates, a reversible QCA half-adder, a reversible QCA half-subtractor, and a reversible QCA 4-to-1 multiplexer to create the proposed reversible QCA AU. The circuit implementation requires 463 QCA cells and 0.83 m^2^ of area, whereas the delay time is 14 clock zones, which equals 3.5 clock cycles.

For the development of reversible QCA CU, the reversible QCA 2-to-1 multiplexer, represented in Figure 19, was used. The CU is crucial for selecting the ALU function, i.e., either an arithmetic or logical operation.

Finally, by combining the layout configurations of the three components LU, AU, and CU, and putting in the required QCA wiring lines, the novel reversible QCA ALU was completed, as illustrated in Figure 23. The proposed reversible QCA ALU implementation requires 1153 QCA cells and costs 2.14 µm^2^ of area. The delay time of this ALU is 24 clock zones, which is equivalent to 6 clock cycles.

## 5. Energy Dissipation Simulation Results and Discussion

The most significant advantage of designing digital circuits to be logically and physically reversible is the improvement of energy efficiency. Therefore, a thorough investigation of the energy dissipation for the proposed logically and physically reversible QCA ALU was conducted in this study. The energy dissipation was calculated for each component of the proposed logically and physically reversible QCA ALU, including the reversible AND, OR, and XOR gates, as well as for the reversible half-adder, half-subtractor, 2-to-1 multiplexer, and 4-to-1 multiplexer circuits. In addition, the energy dissipation of the logically and physically reversible QCA LU, AU, and ALU was evaluated. The energy dissipation values were calculated using the QCADesigner-E simulation tool, and the results are summarized in Table 3. The simulation results demonstrate significant energy efficiency achieved when designing QCA circuits using the physically and logically reversible design technique.

At a temperature of 1 K, every component, including the reversible AND, OR, and XOR gates, as well as the reversible half-adder, half-subtractor, 2-to-1 multiplexer, and 4-to-1 multiplexer circuits, exhibited exceptional energy dissipation values below the Landauer energy limit of *k_B_T*ln2. Moreover, the proposed designs for a logically and physically reversible QCA LU, AU, and ALU possess ultralow energy dissipation with averages of 0.397 meV, 0.405 meV, and 0.908 meV per operation, respectively.

In Table 4, we compare the energy efficiency, number of operations, occupied area, number of required QCA cells, and latency of our logically and physically reversible QCA ALU design to the most recent QCA ALU designs presented in the literature. Additionally, this table presents the method used to deal with wire junctions, as well as the reversibility status of each design. The logically and physically reversible QCA ALU design proposed in this study requires 1153 QCA cells, 2.14 µm^2^ of area, and six clock cycles of delay to execute 16 operations. Using three distinct layers, the multilayer crossover method was utilized for wire crossing.

According to the simulation results presented in Table 4, our proposed logically and physically reversible QCA ALU shows a significant reduction in energy dissipation compared with previous QCA ALU designs. Figure 24 shows that the proposed QCA ALU consumes 88.8% less energy than the most energy-efficient QCA ALU design proposed previously [37].

Figure 25 and Figure 26 demonstrate the number of operations, delay time, occupied area, and required QCA cells for the novel logically and physically reversible QCA ALU design and the existing designs. Although many previous designs utilized fewer QCA cells and shorter delay times [14,16,32,37,38,41,42,43,44], and some others occupied less area [16,32,37,41,42,43,44] than the proposed ALU, these ALUs performed fewer operations. These previous ALUs can perform either 12 operations [42], 10 operations [38], eight operations [14,37], or just four operations [16,32,41,43,44].

Thus, for a more fair comparison, we compared our proposed ALU, which can perform 16 operations, with QCA ALU designs that can perform a similar number of operations [13,39,40], as presented in Figure 27 and Figure 28. This comparison demonstrates that the logically and physically reversible design proposed in this study requires 51% fewer QCA cells, 47% less area, and a comparable latency compared to the best QCA ALU design previously presented that can perform 16 operations [40].

## 6. Conclusions

The present work introduces a brand-new multilayer design of a logically and physically reversible QCA ALU with exceptionally low energy dissipation values, two orders of magnitude lower than other designs in scientific literature. The fact that reversibility is maintained down to the layout level, which is the physical representation of the circuit, is the major advantage of this design. Theoretically, this means no information is being lost, and consequently, the amount of energy dissipated into the environment is zero. In this research, we developed a building block majority gate that is both logically and physically reversible to create all components of a QCA ALU with very low power consumption. The multilayer crossover method was used to prevent wire crosstalk. The USE clocking scheme was used to synchronize the data flow accurately and guarantee the correct operation of the ALU. The reversible QCA ALU presented here can perform sixteen distinct operations, half of which are logical and the other half arithmetic. The simulation of performance and the evaluation of energy dissipation were carried out using the *QCADesigner-E* program. To implement 16 operations, the logically and physically reversible QCA ALU design presented in this study employs 1153 QCA cells, uses an area of 2.14 m2, and has a delay of 6 clock cycles.

The simulation results confirm that the concept of logically and physically reversible design can lead to circuits that lose almost no energy during operation. Each logically and physically reversible gate/circuit that was used to build up the proposed ALU, such as the reversible AND, OR, and XOR gates and the reversible half-adder, half-subtractor, 2-to-1 multiplexer, and 4-to-1 multiplexer circuits, had energy dissipation values below the Landauer energy limit of *k_B_T*ln2. Additionally, the findings from the simulation demonstrated that the proposed logically and physically reversible QCA ALU showed an improvement in energy efficiency by 88.8% compared with the recent design of M. Patidar et al. J. Supercomput. 2023 [37]. In addition, when compared to the most efficient 16-operation QCA ALU designs that were previously presented, this ALU design utilizes 51% fewer QCA cells and 47% less area than the other designs.

In the future, the current proposed logically and physically reversible QCA ALU design can be extended to handle more operations. In addition, it can be expanded to the situation of ultra-energy-efficient multiple-bit ALU circuits. There is exciting recent experimental work where the QCA quantum dot building blocks consist of silicon dangling bonds on a Hydrogen terminated silicon surface [45,46]. Artificial molecules can be precisely assembled from these quantum dots and can be placed on the surface to act as QCA-based digital logic circuit elements, whose operation has been verified experimentally. The overarching aim of the present simulation work is to inform these state-of-the-art sophisticated nanoscience experiments to push the area of QCA forward.

## Figures and Tables

**Figure 1 nanomaterials-13-02445-f001:**
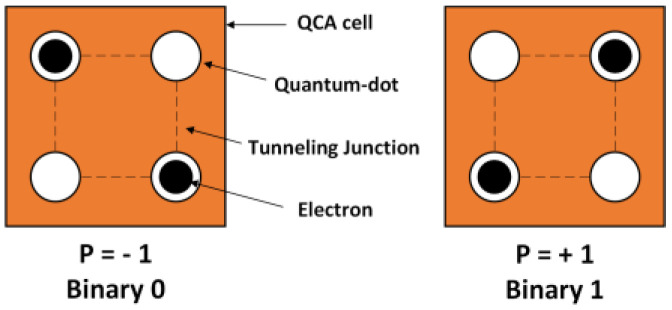
QCA cell polarization.

**Figure 2 nanomaterials-13-02445-f002:**
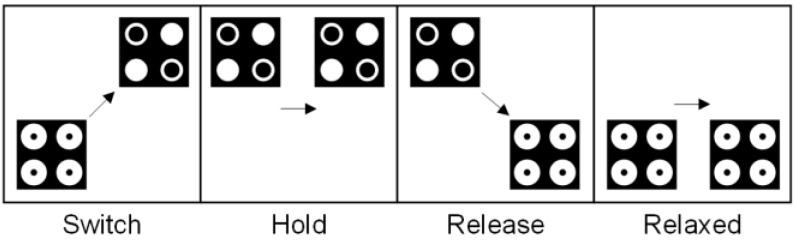
Adiabatic pipelining phases (the arrows are used to illustrate the change in polarization state of the QCA cell during the four stages of the clock cycle).

**Figure 3 nanomaterials-13-02445-f003:**
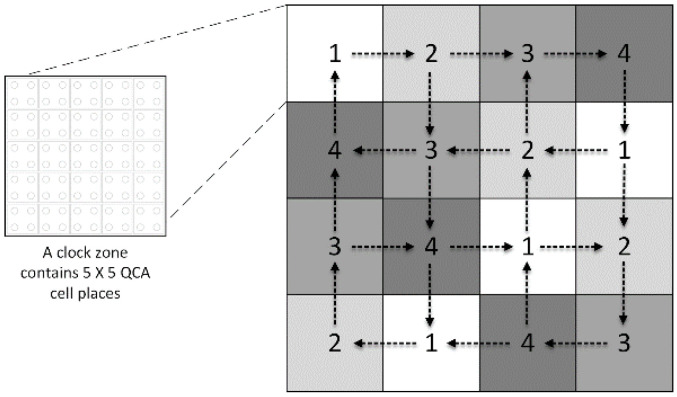
The USE timing mechanism (the squares are used to represent time zones, and the arrows are used to represent data flow).

**Figure 4 nanomaterials-13-02445-f004:**
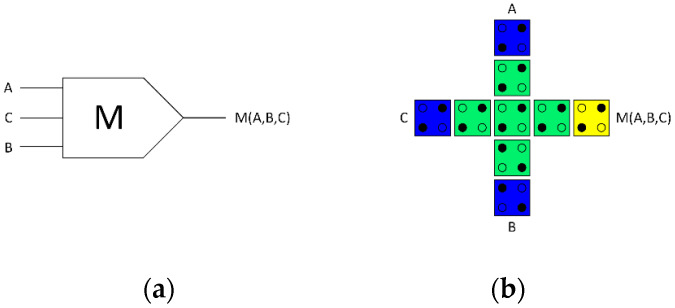
(**a**) Logical synthesis of the standard irreversible majority gate, (**b**) physical layout of the standard irreversible QCA majority gate.

**Figure 5 nanomaterials-13-02445-f005:**
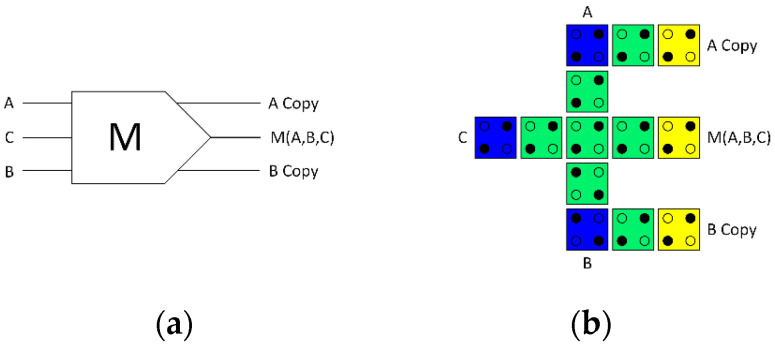
(**a**) Logical synthesis of the reversible majority gate, (**b**) physical layout of the reversible QCA majority gate.

**Figure 6 nanomaterials-13-02445-f006:**
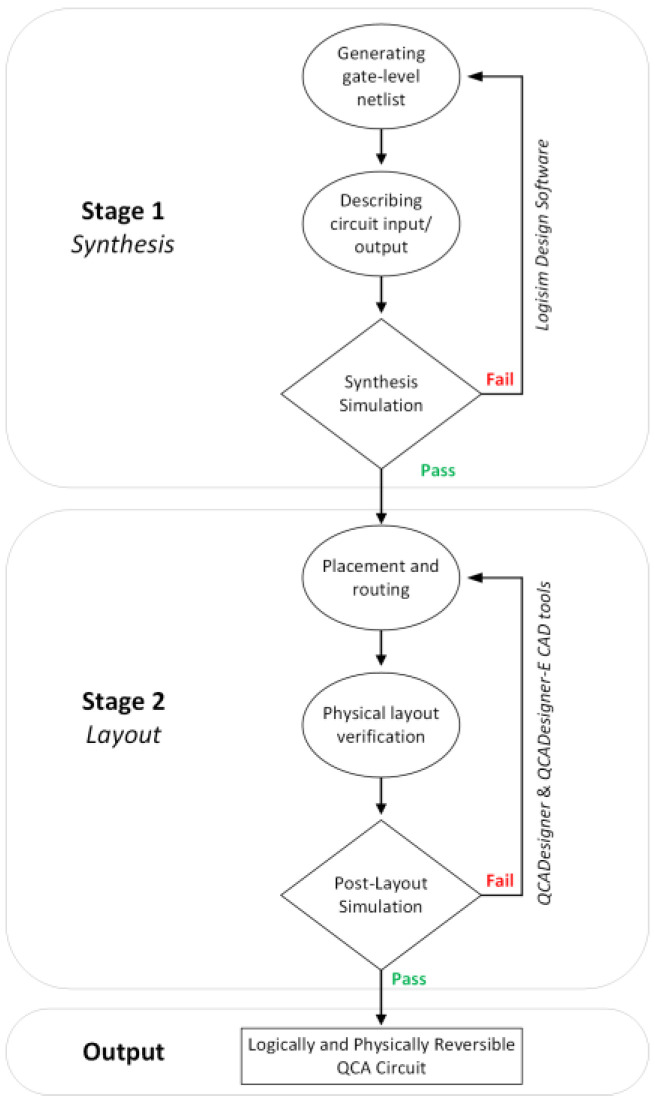
The methodology of designing logically and physically reversible QCA circuits.

**Figure 7 nanomaterials-13-02445-f007:**
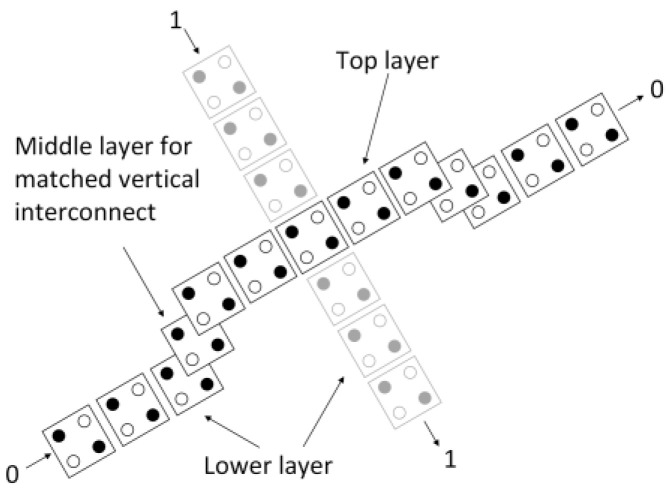
Multilayer crossover method (the black and grey colorings of the two lines represent the two distinct wires that are crossing).

**Figure 8 nanomaterials-13-02445-f008:**
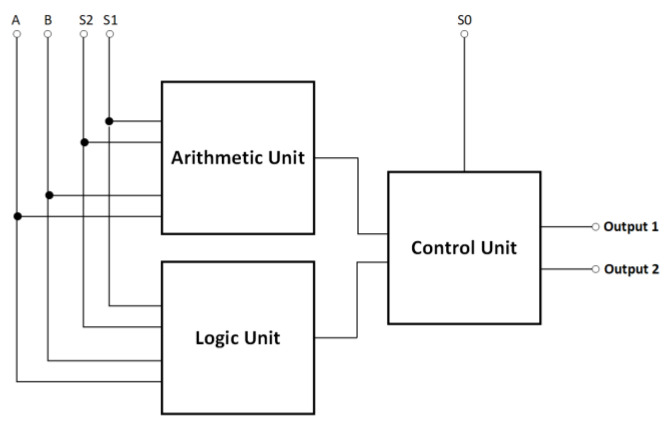
The High-level block diagram of the proposed reversible QCA ALU.

**Figure 9 nanomaterials-13-02445-f009:**
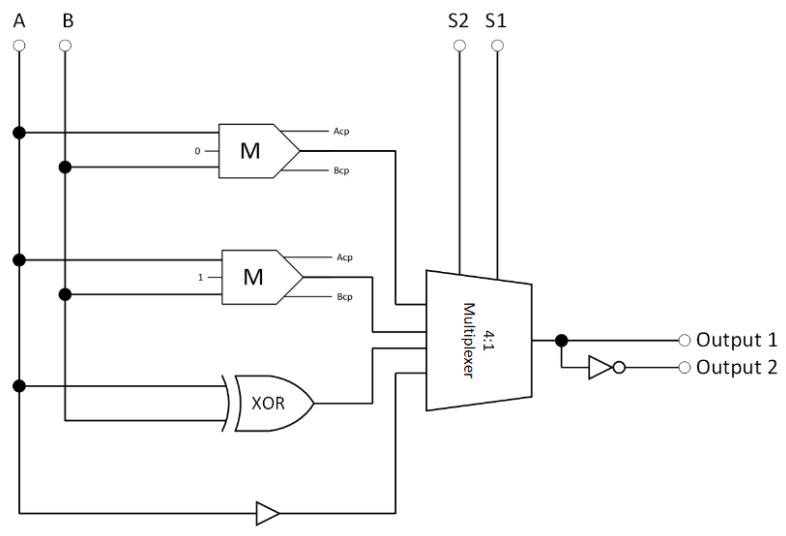
The synthesis of the proposed reversible LU (A_cp_ and B_cp_ refer to copies of the inputs).

**Figure 10 nanomaterials-13-02445-f010:**
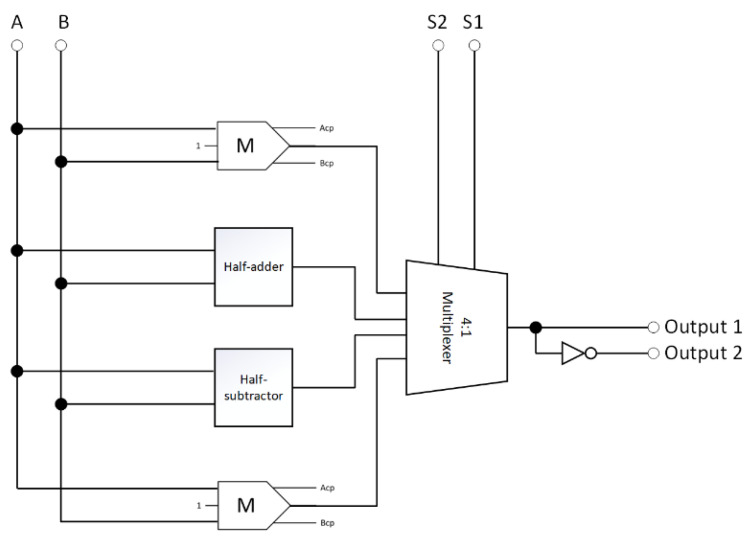
The synthesis of the proposed reversible AU (A_cp_ and B_cp_ refer to copies of the inputs).

**Figure 11 nanomaterials-13-02445-f011:**
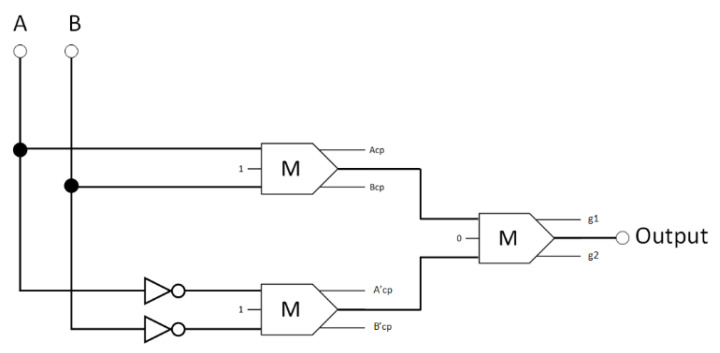
The synthesis of the proposed reversible XOR (A_cp_, B_cp_, A′_cp_, and B′_cp_ refer to copies of the input data, whereas g1 and g2 indicate the garbage outputs).

**Figure 12 nanomaterials-13-02445-f012:**
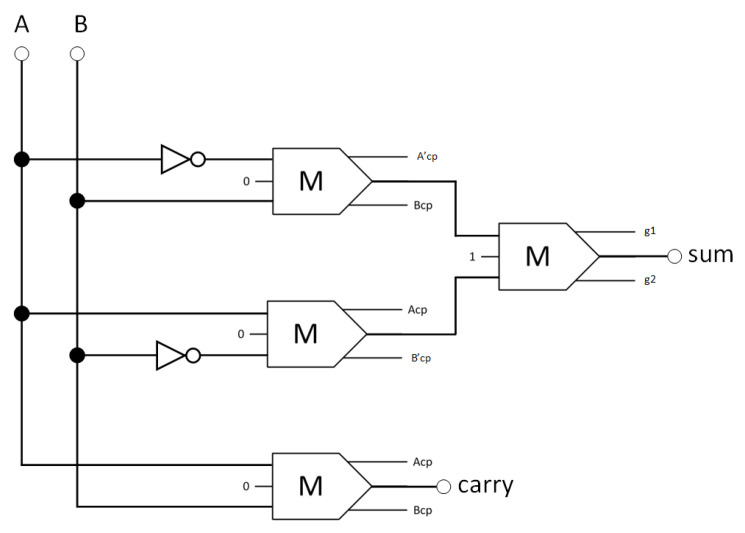
The synthesis of the proposed reversible half-adder (A_cp_, B_cp_, A′_cp_, and B′_cp_ refer to copies of the input data, whereas g1 and g2 indicate the garbage outputs).

**Figure 13 nanomaterials-13-02445-f013:**
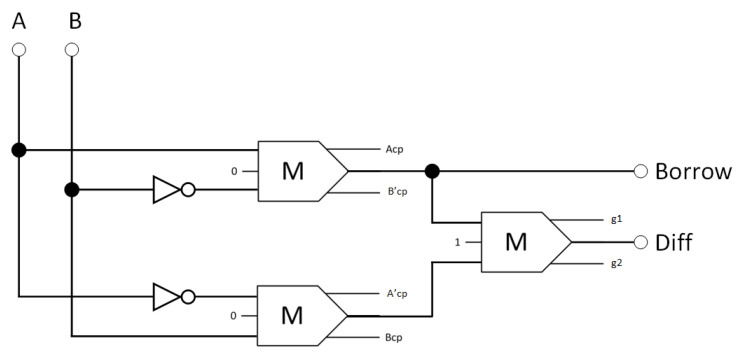
The synthesis of the proposed reversible half-subtractor (A_cp_, B_cp_, A′_cp_, and B′_cp_ refer to copies of the input data, whereas g1 and g2 indicate the garbage outputs).

**Figure 14 nanomaterials-13-02445-f014:**
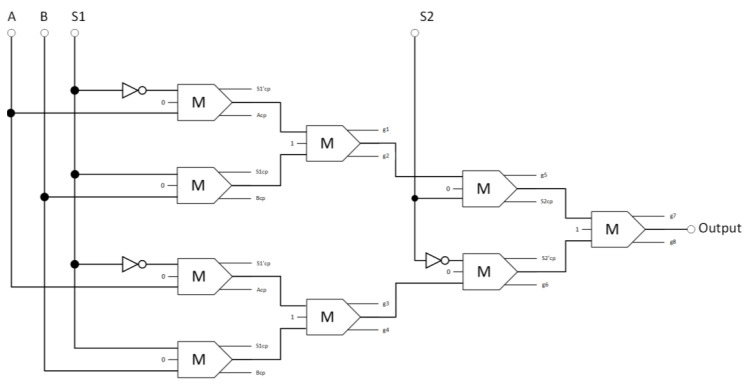
The synthesis of the proposed reversible 4:1 multiplexer (A_cp_, B_cp_, S1_cp_, S2_cp_, S1′_cp_, and S2′_cp_ refer to copies of the input data, whereas g1 and g2 indicate the garbage outputs).

**Figure 15 nanomaterials-13-02445-f015:**
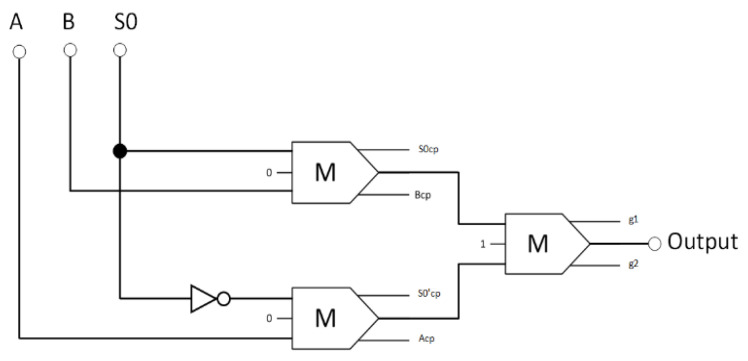
The synthesis of the proposed reversible 2:1 multiplexer (A_cp_, B_cp_, S0_cp_, and S0′_cp_ refer to copies of the input data, whereas g1 and g2 indicate the garbage outputs).

**Figure 16 nanomaterials-13-02445-f016:**
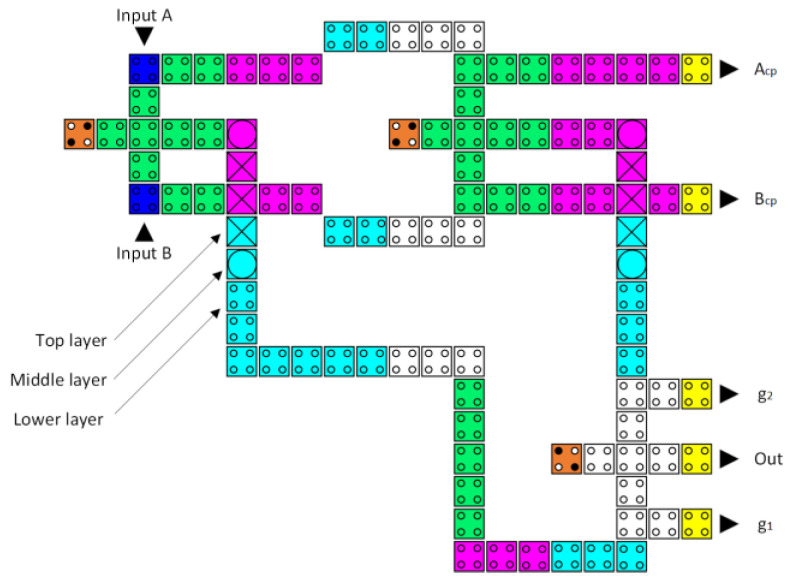
The layout of the proposed reversible QCA XOR (A_cp_ and B_cp_ refer to copies of the input data, whereas g1 and g2 indicate the garbage outputs).

**Figure 17 nanomaterials-13-02445-f017:**
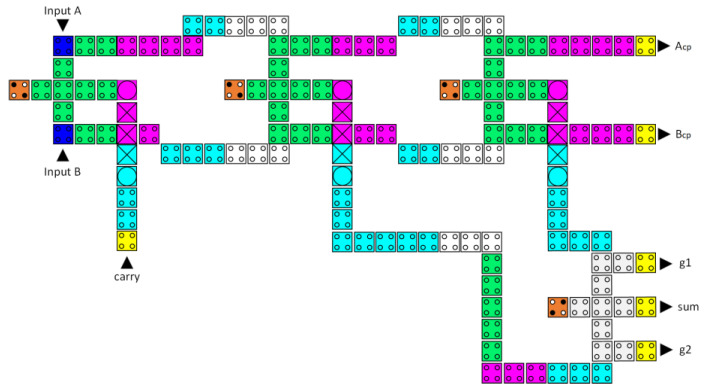
The layout of the proposed reversible QCA half-adder (A_cp_ and B_cp_ refer to copies of the input data, whereas g1 and g2 indicate the garbage outputs).

**Figure 18 nanomaterials-13-02445-f018:**
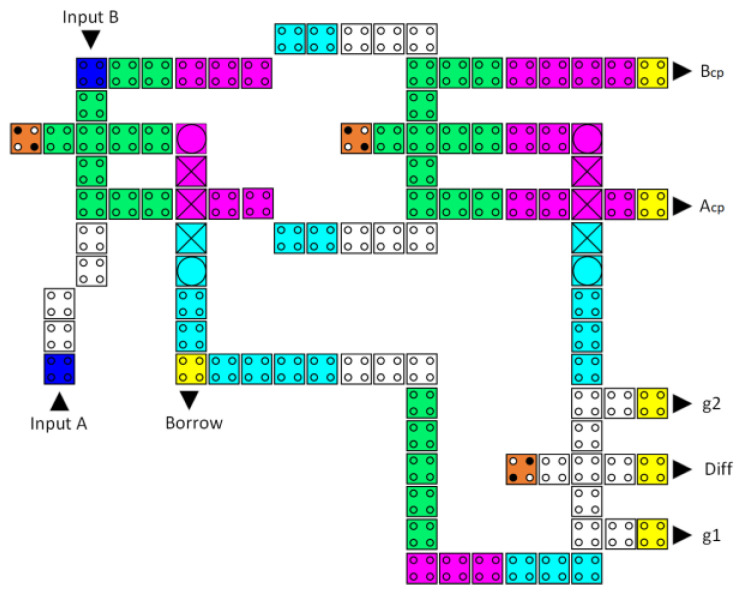
The layout of the proposed reversible QCA half-subtrator (A_cp_ and B_cp_ refer to copies of the input data, whereas g1 and g2 indicate the garbage outputs).

**Figure 19 nanomaterials-13-02445-f019:**
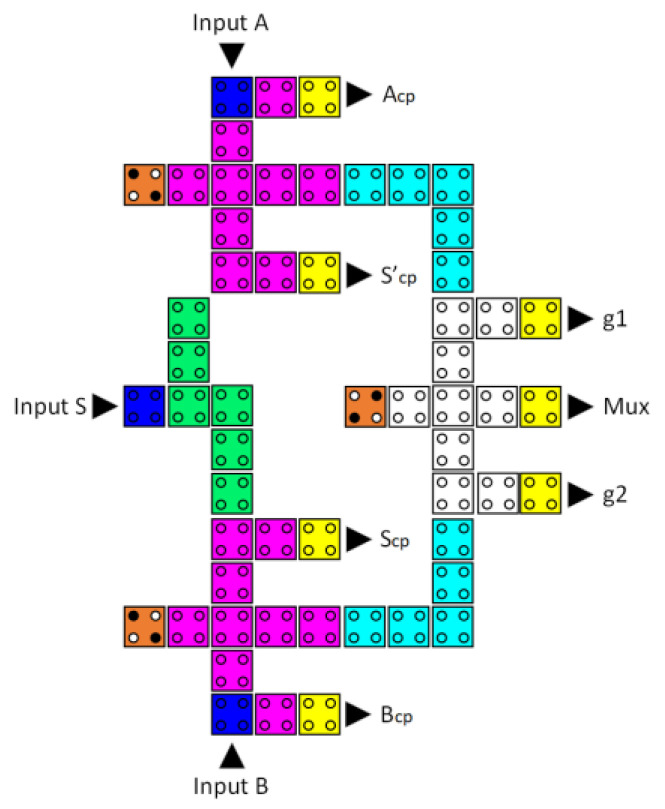
The layout of the proposed reversible QCA 2-to-1 multiplexer (A_cp_, B_cp_, S_cp_, and S’_cp_ refer to copies of the input data, whereas g1 and g2 indicate the garbage outputs).

**Figure 20 nanomaterials-13-02445-f020:**
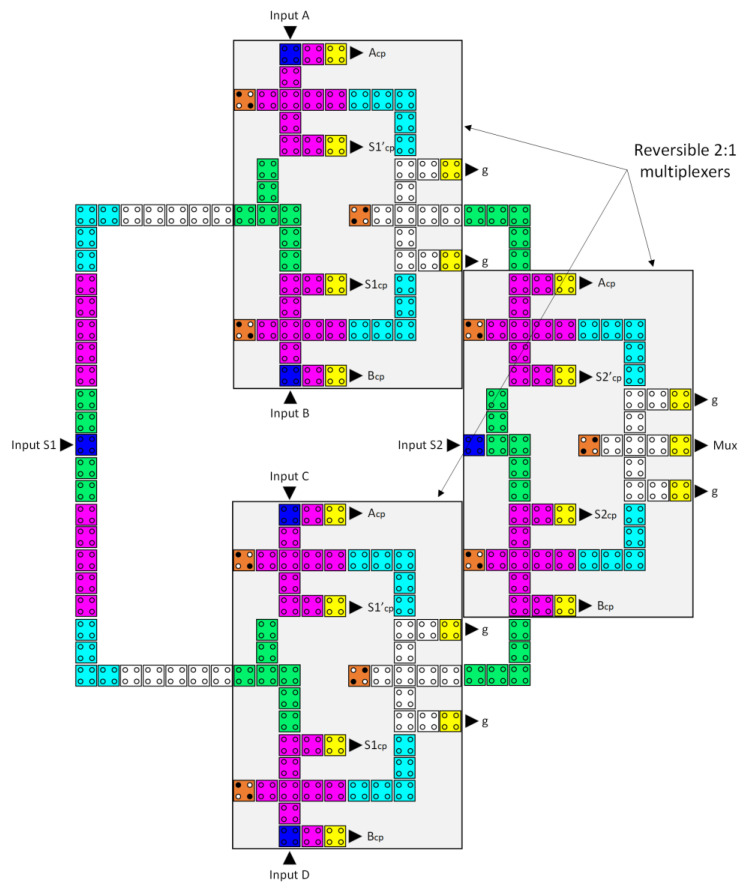
The layout of the proposed reversible QCA 4-to-1 multiplexer (A_cp_, B_cp_, C_cp_, D_cp_, S1_cp_, S1′_cp_, S2_cp_, and S2′cp refer to copies of the input data, whereas g variables indicate the garbage outputs).

**Figure 21 nanomaterials-13-02445-f021:**
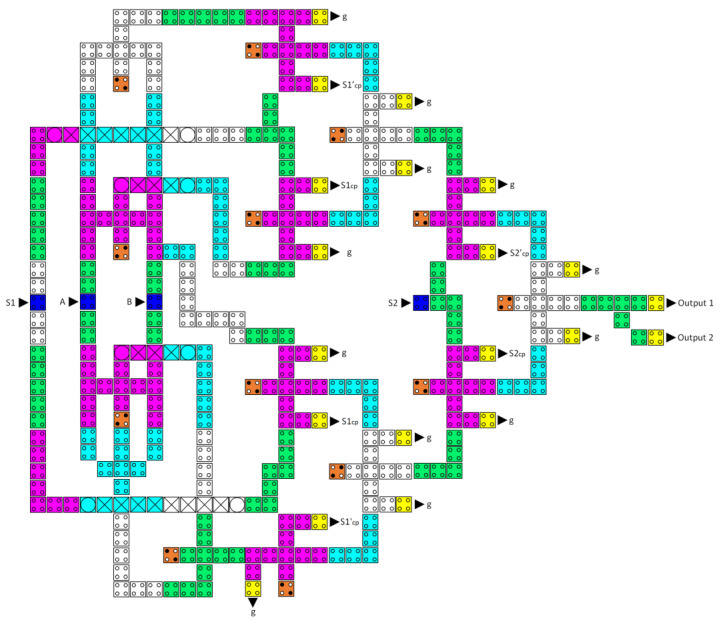
The layout of the proposed reversible QCA LU (S1_cp_, S1′_cp_, S2_cp_, and S2′_cp_ refer to copies of the input data, whereas g variables indicate the garbage outputs).

**Figure 22 nanomaterials-13-02445-f022:**
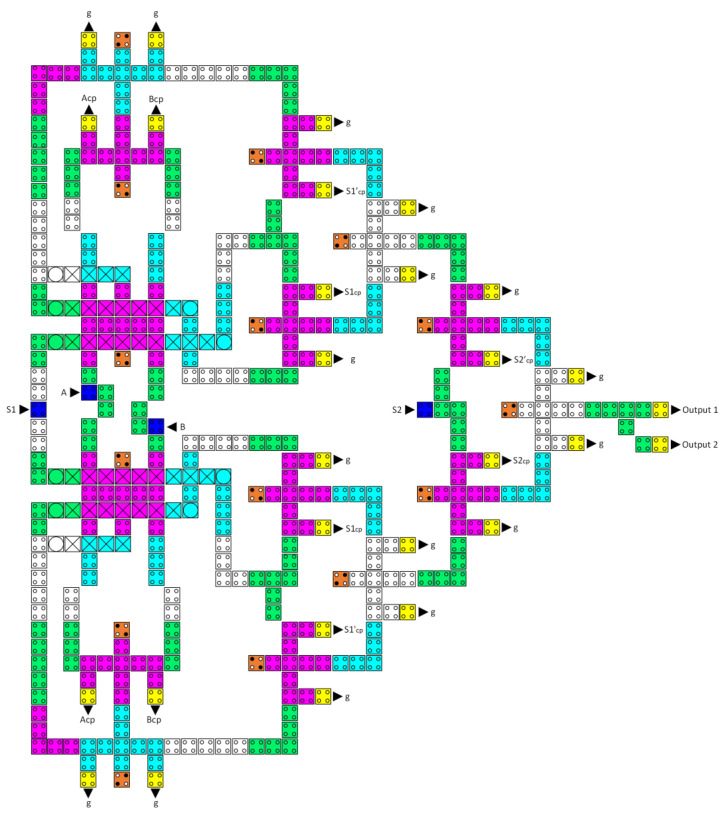
The layout of the proposed reversible QCA AU (A_cp_, B_cp_, S1_cp_, S1′_cp_, S2_cp_, and S2′_cp_ refer to copies of the input data, whereas g variables indicate the garbage outputs).

**Figure 23 nanomaterials-13-02445-f023:**
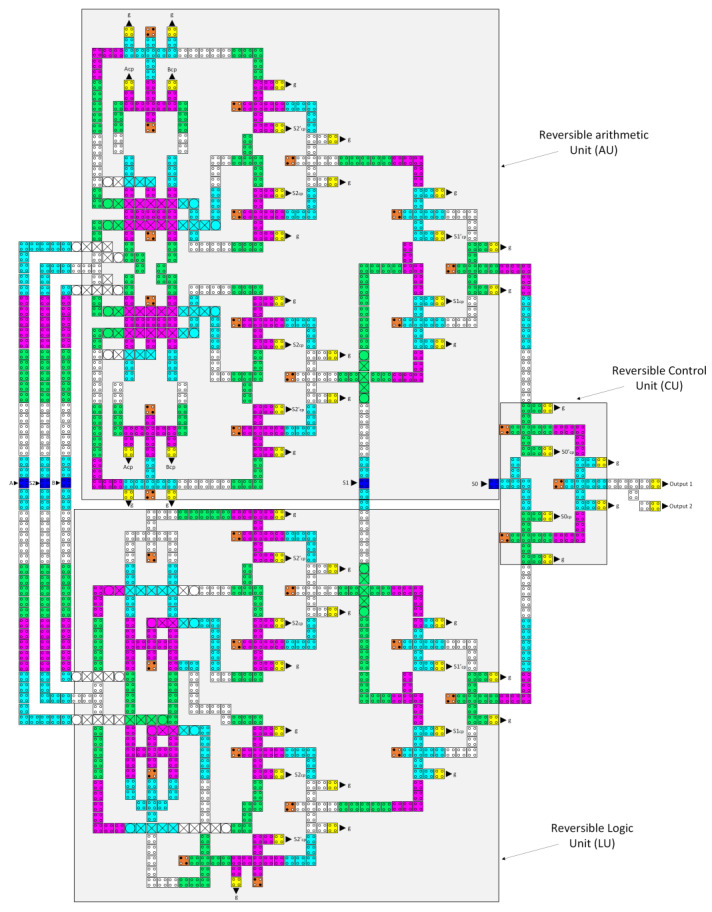
The layout of the proposed reversible QCA ALU (A_cp_, B_cp_, C_cp_, D_cp_, S1_cp_, S1′_cp_, S1_cp_, S1′_cp_, S2_cp_, and S2′_cp_ refer to copies of the input data, whereas g variables indicate the garbage outputs).

**Figure 24 nanomaterials-13-02445-f024:**
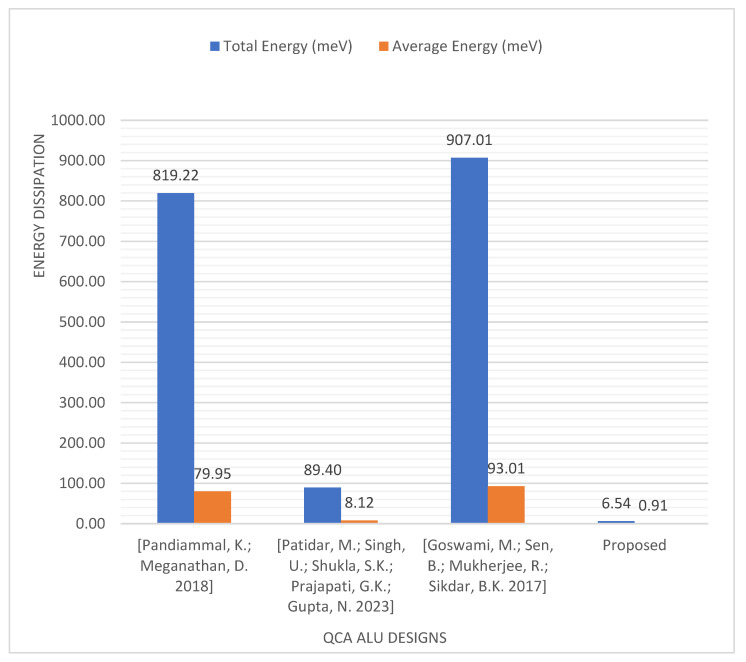
Energy dissipation comparison of QCA ALU designs [32,37,38].

**Figure 25 nanomaterials-13-02445-f025:**
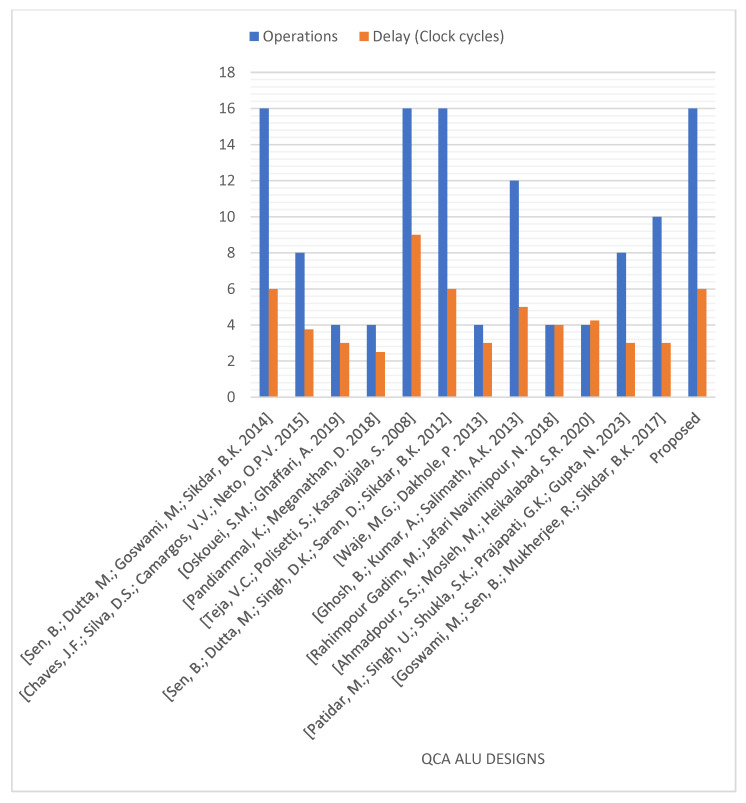
Number of operations and delay time of the QCA ALU designs [13,14,16,32,37,38,39,40,41,42,43,44].

**Figure 26 nanomaterials-13-02445-f026:**
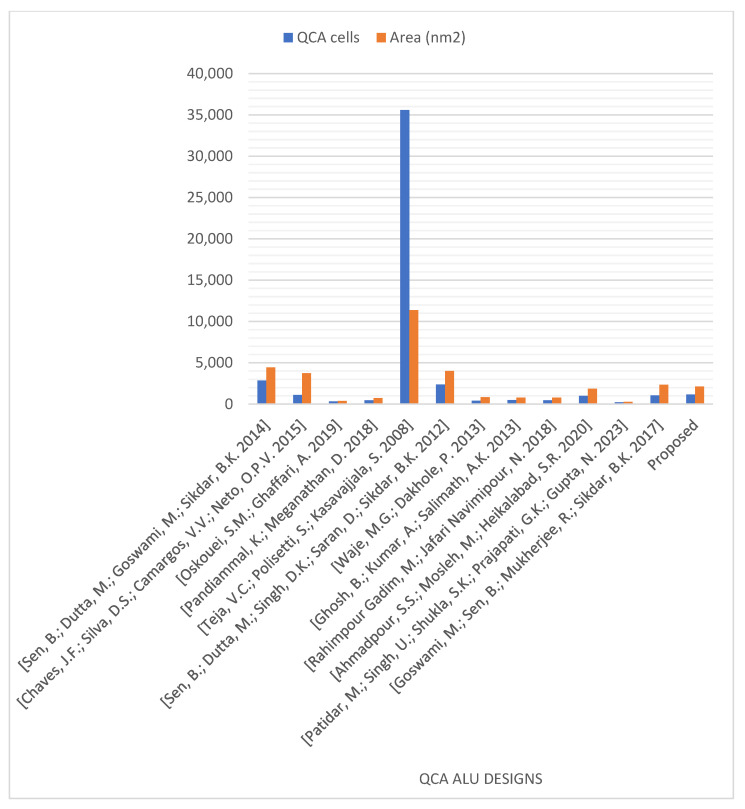
QCA cells and occupied areas were utilized to design the QCA ALUs [13,14,16,32,37,38,39,40,41,42,43,44].

**Figure 27 nanomaterials-13-02445-f027:**
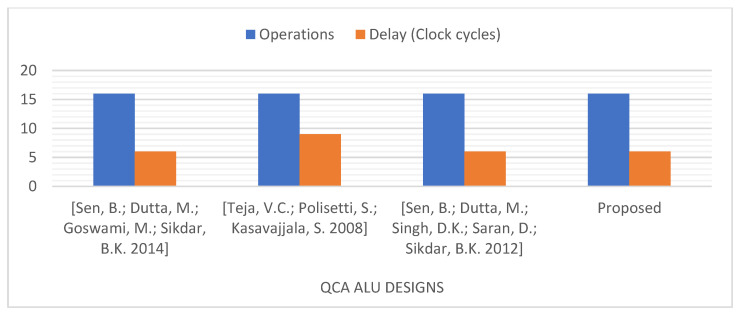
Number of operations and delay time of the QCA ALU designs that perform 16 operations [13,39,40].

**Figure 28 nanomaterials-13-02445-f028:**
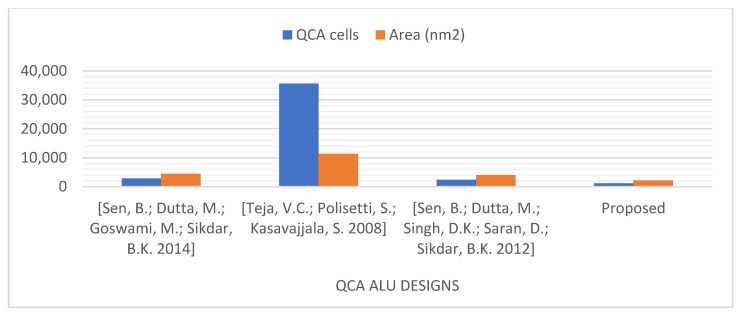
QCA cells and occupied areas are utilized to design the QCA ALUs that can perform 16 operations [13,39,40].

**Table 1 nanomaterials-13-02445-t001:** The employed technology and simulation parameters.

Parameter	Description	Value
QD size	Quantum-dot size	5 nm
Cell area	Dimensions of each cell	18 × 18 nm
Cell distance	Distance between two cells	2 nm
Layer separation	Distance between QCA layers in multilayer crossing	11.5 nm
Clock high	Max. saturation energy of clock signal	9.8 × 10^−22^ J
Clock low	Min. saturation energy of clock signal	3.8× 10^−23^ J
Relative permittivity	Relative permittivity of material for QCA system (GaAs & AlGaAs)	12.9
Radius of effect	Maximum distance between cells whose interaction is considered	80 nm
Temp	Operating temperature	1 K
*τ*	Relaxation time	1 × 10^−15^ s
*T_γ_*	Period of the clock signal	1 × 10^−9^ s
*T_in_*	Period of the input signals	1 × 10^−9^ s
*T_step_*	Time interval of each iteration step	1 × 10^−16^ s
*T_sim_*	Total simulation time	8 × 10^−9^ s
*γ_shape_*	Shape of clock signal slopes	GAUSSIAN
*γ_slope_*	Rise and fall time of the clock signal slopes	1 × 10^−10^ s

**Table 2 nanomaterials-13-02445-t002:** The operations of the proposed reversible QCA ALU.

Operation Type	Control Inputs			Output 1	Output 2 (Inversion of Output 1)
	S_0_	S_1_	S_2_		
Logic operations (LU)	0	0	0	AND	NAND
	0	0	1	OR	NOR
	0	1	0	Buffer	NOT (Inverter)
	0	1	1	XOR	XNOR
Arithmetic operations (AU)	1	0	0	A + B	1′Complement (A + B)′
	1	0	1	C_out_	A.B
	1	1	0	A′·B	(A′B)′
	1	1	1	A − B	(A − B)′

**Table 3 nanomaterials-13-02445-t003:** The energy dissipation analysis of the proposed reversible QCA ALU. Note that the average energy dissipation denotes the mean energy value averaged over the various input signal combinations.

Logically and Physically Reversible QCA ALU Build-Up Circuits	Total Energy Dissipation (meV)	Average Energy Dissipation (meV)
Reversible AND	0.009	0.002
Reversible OR	0.009	0.002
Reversible XOR	0.054	0.014
Reversible half-adder	0.099	0.025
Reversible half-subtractor	0.063	0.016
Reversible 4:1 multiplexer	0.525	0.057
Reversible CU (Reversible 2:1 multiplexer)	0.112	0.014
Reversible LU	2.28	0.397
Reversible AU	2.84	0.405
Reversible ALU	6.54	0.908

**Table 4 nanomaterials-13-02445-t004:** Comparison of performance and energy dissipation. Note that there are only three references [32,37,38] that calculate the energy dissipation for the QCA ALU.

Reference	Operations	QCA Cells	Area (nm^2^)	Delay (Clock Cycles)	Wire Crossing	Total Energy Dissipation (meV)	Average Energy Dissipation (meV)	Reversibility
[13]	16	2857	4440	6	Coplanar	NG	NG	Logically
[14]	8	1097	3740	3.75	Multilayer	NG	NG	Logically
[16]	4	332	380	3	Multilayer	NG	NG	Logically
[32]	4	452	740	2.5	Coplanar	819.22	79.95	Irreversible
[39]	16	35,596	11,370	9	Coplanar	NG	NG	Irreversible
[40]	16	2370	4010	6	Coplanar	NG	NG	Logically
[41]	4	420	850	3	Multilayer	NG	NG	Irreversible
[42]	12	485	790	5	Multilayer	NG	NG	Irreversible
[43]	4	464	780	4	Multilayer	NG	NG	Irreversible
[44]	4	1010	1860	4.25	Coplanar	NG	NG	Irreversible
[37]	8	231	280	3	Multilayer	89.40	8.12	Irreversible
[38]	10	1069	2340	3	Coplanar	907.01	93.01	Logically
Proposed	16	1153	2140	6	Multilayer	6.54	0.908	Logically & physically

## Data Availability

Not applicable.

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
