# Peer review of "Reversible Quantum-Dot Cellular Automata-Based Arithmetic Logic Unit"

_nanomaterials, 2023, doi:10.3390/nano13172445_

Round 1
Reviewer 1 Report
17 Aug. 2023 - nanomaterials-2546096-report on
Reversible Quantum-Dot Cellular Automata-Based Arithmetic Logic Unit
by Mohammed Alharbi, Gerrard Edwards, and Richard Stocker
The paper discusses the application of quantum-dot cellular automata (QCA) to improve energy dissipation in future arithmetic logic units (ALUs). QCA can achieve higher speed, lower power, and smaller areas than conventional metal-oxide semiconductor (CMOS) technology. Reversible QCA design can provide exceptional reductions in energy dissipation. Current QCA ALU designs are either irreversible or logically reversible, but they lack physical reversibility, a crucial requirement for increased energy efficiency. This paper presents a new multilayer design for a QCA ALU that is both logically and physically reversible, and offers an improvement of 88.8% in energy efficiency.
The paper is well written and do demonstrate a path towards designing reversible components of a QCA ALU by showing simulation results and comparison with the currently known alternatives.
I have only a few minor suggestions:
1) Explain the sub-index 'i' on H in (1);
2) Regarding section 3. Energy Behavior of QCA Cells.
2.a) Explain in more details how the 2x2 Pauli sigmas in the Hamiltonian relates to the 3D coherence vector λ and energy vector Γ.
2.b) Elaborate more on how the energy conservation in (11) is retained. That is, which components are negative and which are positive, and when and how they cancel to zero.
3) Putting the legends in the Figures 24 to 28 within the bar-char space and not on the side as done at present.
4) Adding a notation table related to the various notations used in the formulas and the circuit designs.
Reviewer 2 Report
Report on:
Reversible Quantum-Dot Cellular Automata-Based Arithmetic Logic Unit
by Mohammed Alharbi*Xueyu Chen at al
The paper describes the model of an arithmetic logic unit built on a quantum-dots approach. The topic is heavily discussed in the community and the research has some potential to attract reader’ interest. Nevertheless, I have two major concerns and few small comments to the contents of the presented study.
Concerns:
Presentation: the paper is too big in size for the quality of material presented. One can broadly imagine two types of a scientific paper: report or review. Current manuscript is approaching a review in size, but essential material could be easily fit into a few pages report. For example, Section 3 can be completely removed without any damage to the paper integrity. Moreover, Section 4 (“methodology”) is full of diagrams, but doesn’t present any significant description of the methods used in the research. So, I encourage authors to regroup the material, shorted the contents and highlight their achievements.
Implementation: The presented results are fully simulation-based. One can understand the technological challenges in this domain, but I do miss a discussion of the technical feasibility of the proposed design. That becomes particularly tricky in Section 6 where the authors compare their design to others and conclude about excellency. One rather straightforward answer to these considerations could be that the other groups present a ‘real-world’ device, while the authors – only a ‘dream’.
References: the style of bibliography list is changing from item to item. Please, fix. All citations are relatively old, the only two quite recent publications are self-citations. Please, provide a more up-to-date bibliography
Detailed comments:
L16: QCA is not defined
L19-20: challenge of reversible QCD…reversibility : it looks like a circular argument
L66: polarity of the electron orientation polarity : what is that?
L74: as the polarization of electrons: do you mean spin?
L89: newly proposed reversible gates: provide a citation
L245,249: can provide, can describe : one should not use “can” in this context.
Figure 4: I can’t get the method from this diagram. Is that: in case of Failure repeat again?
Figure 7,13,18: do not split the picture and the caption.
L520: the exceptional energy efficiency: this statement is rather arbitrary. I would not use such a strong word.
Table 3: average energy dissipations is not defined
Table 4: I am not sure what to conclude from these numbers. Is “proposed” the best or simply unrealistic? Or simulation has a bug?
In summary, I think the paper must be improved before it can be published.
